# Deregulated E2F Activity as a Cancer-Cell Specific Therapeutic Tool

**DOI:** 10.3390/genes14020393

**Published:** 2023-02-02

**Authors:** Rinka Nakajima, Lin Zhao, Yaxuan Zhou, Mashiro Shirasawa, Ayato Uchida, Hikaru Murakawa, Mariana Fikriyanti, Ritsuko Iwanaga, Andrew P. Bradford, Keigo Araki, Tomoko Warita, Kiyoshi Ohtani

**Affiliations:** 1Department of Biomedical Sciences, School of Biological and Environmental Sciences, Kwansei Gakuin University, 1 Gakuen Uegahara, Sanda, Hyogo 669-1337, Japan; 2Department of Obstetrics and Gynecology, University of Colorado School of Medicine, Anschutz Medical Campus, 12800 East 19th Avenue, Aurora, CO 80045, USA; 3Department of Morphological Biology, Ohu University School of Dentistry, 31-1 Misumido Tomitamachi, Koriyama, Fukushima 963-8611, Japan

**Keywords:** E2F, pRB, p53, ARF

## Abstract

The transcription factor E2F, the principal target of the tumor suppressor pRB, plays crucial roles in cell proliferation and tumor suppression. In almost all cancers, pRB function is disabled, and E2F activity is enhanced. To specifically target cancer cells, trials have been undertaken to suppress enhanced E2F activity to restrain cell proliferation or selectively kill cancer cells, utilizing enhanced E2F activity. However, these approaches may also impact normal growing cells, since growth stimulation also inactivates pRB and enhances E2F activity. E2F activated upon the loss of pRB control (deregulated E2F) activates tumor suppressor genes, which are not activated by E2F induced by growth stimulation, inducing cellular senescence or apoptosis to protect cells from tumorigenesis. Deregulated E2F activity is tolerated in cancer cells due to inactivation of the ARF-p53 pathway, thus representing a feature unique to cancer cells. Deregulated E2F activity, which activates tumor suppressor genes, is distinct from enhanced E2F activity, which activates growth-related genes, in that deregulated E2F activity does not depend on the heterodimeric partner DP. Indeed, the ARF promoter, which is specifically activated by deregulated E2F, showed higher cancer-cell specific activity, compared to the E2F1 promoter, which is also activated by E2F induced by growth stimulation. Thus, deregulated E2F activity is an attractive potential therapeutic tool to specifically target cancer cells.

## 1. Introduction

Cancer is a major cause of death in many countries, and its incidence is increasing worldwide. Surgical treatment is the first choice, but it is not efficacious in metastatic tumors or leukemia. In such cases, radiation or chemotherapy is applied; however, in many cases, these treatments have limited effects on disease progression and overall survival, and side effects caused by these therapies remain a major problem in cancer treatment. These therapies preferentially target proliferating cells, including cancer cells, but also damage normal growing cells, causing side effects which hamper the radical treatment of cancer. To avoid such side effects, we need to identify potential molecular mechanisms that discriminate cancer cells from normal growing cells, thereby facilitating the specific targeting of malignancies. Many approaches are being investigated based on this premise, including E2F-based strategies.

The transcription factor E2F is the principal target of the tumor suppressor pRB, and it plays pivotal roles in cell growth, apoptosis, and other biological processes [1,2,3,4,5,6,7]. In resting normal cells, the activity of E2F is suppressed by the binding of pRB and its family members p107 and p130 (collectively referred to as RB). Upon growth stimulation, RB is inactivated by cyclin-dependent kinases, leading to the activation of E2F, which induces growth-related gene expression and promotes cell proliferation. Cancer cells are characterized by unrestrained cell growth mediated by enhanced E2F activity due to the loss of pRB function caused by a variety of oncogenic changes. Based on this fact, approaches are undertaken to either restrain cell growth, by suppressing enhanced E2F activity, or conversely, to induce cytotoxic responses in cancer cells, utilizing enhanced E2F activity. However, these strategies may also suppress growth or adversely impact the viability of normal growing cells.

E2F also plays crucial roles in tumor suppression by inducing apoptosis [1,2,3,4,5,6,7]. In almost all cancers, pRB function is disabled by oncogenic changes resulting in increased, unrestrained E2F activation, no longer subject to control by pRB [8]. Such deregulated E2F activity stimulates the transcription of pro-apoptotic tumor suppressor genes and induces apoptosis, thereby protecting cells from tumorigenesis. Notably, E2F activity induced by growth stimulation does not activate these tumor suppressor genes [9,10,11]. Hence, deregulated E2F activity that activates tumor suppressor genes is not present in normal growing cells. In contrast, cancer cells survive by accumulating mutations in apoptosis-inducing pathways, hence deregulated E2F activity exists specifically in cancer cells. Thus, it is expected that utilizing deregulated E2F activity will facilitate the specific targeting of cancer cells, while sparing and preserving normal growing cells.

Since E2F plays crucial roles in cell cycle progression, cell proliferation, tumorigenesis, apoptosis, and tumor suppression, there are comprehensive studies from other laboratories on these topics [1,2,3,4,5,7,12,13,14,15]. In this review, we will briefly introduce some of these areas and discuss the therapeutic potential of modulating E2F-dependent proliferation and apoptosis in cancer cells, with a focus on deregulated E2F activity.

## 2. E2 Transcription Factor (E2F)

E2F was identified as a host cell transcription factor, which mediates induction of *E2* gene expression by adenovirus E1a that is expressed immediately after virus infection [16]. Subsequently, E2F was shown to regulate cell proliferation by activating a group of growth-related genes [17]. E2F also plays crucial roles in tumor suppression by activating tumor suppressor genes to induce cellular senescence or apoptosis [1,2,3,4,5,6,7]. E2F was shown to be the principal target of the tumor suppressor pRB and its family members p107 and p130 (collectively referred to as RB) [17,18,19,20]. In the resting state, the activity of E2F is suppressed by the binding of RB.

There are eight E2F family members (E2F1~E2F8), which are divided into activator E2Fs and repressor E2Fs, depending on their primary effects on transcription (Figure 1) [2,3,7]. E2F1~E2F3a are activator E2Fs, whose expression is induced at the G1/S boundary by E2F itself [21,22,23]. At the cell cycle stage, pRB has been inactivated through phosphorylation by cyclin-dependent kinases (CDKs), and activator E2Fs are free from pRB suppression, enabling activation of target genes [18]. Activator E2Fs and E2F3b harbor a nuclear localization signal (NLS) in their N-terminal region and, after being translated in the cytoplasm, undergo translocation into the nucleus. E2F3a and E2F3b are generated by alternative usage of promoters. A recent report describes the identification of new E2F3 members, E2F3c and E2F3d, which are generated by alternative splicing of E2F3a and do not seem to function as transcriptional regulators [24]. Both lack a DNA binding domain and localize in the cytoplasm. E2F3d has a mitochondria targeting signal and was found to mediate hypoxia-induced mitophagy in cancer cells by functioning as a mitophagy receptor [24]. E2F3b~E2F5 repress target gene expression by the binding of the RB family members pRB and p130, and thus are grouped into repressor E2Fs [2,3,7]. E2F4-6 lack an NLS, and E2F4-E2F5 possess a bipartite nuclear export signal (NES). E2F6~E2F8 repress target gene expression independent of RB. E2F1~E2F6 bind to target genes, depending on the heterodimeric partner DP1 or DP2 [25]. E2F7 and E2F8 possess two DNA binding domains and bind to target genes independent of the DP partner [26,27]. There are two isoforms of E2F7, E2F7a and 7b, which are generated by alternative splicing [26].

## 3. Mechanism of Cell Proliferation Mediated by E2F

The proliferation of mammalian cells depends on growth stimulation, which promotes cell cycle progression. Cell cycle progression is governed by CDKs (Figure 2). Progression of the cell cycle to the restriction point located in late G1 depends on growth stimulation. After passing through the restriction point, cell cycle progression proceeds autonomously and no longer depends on growth stimulation. Thus, this restriction point is an important determinant of the cells’ decision to proliferate or remain growth arrested.

The two major regulatory players in the G1 restriction point are pRB and E2F. E2F regulates a group of growth-related genes and plays central roles in progression through the restriction point. In the resting state of the cell cycle, E2F target genes are suppressed by repressor E2Fs bound by RB (Figure 3). The major repressive complex is the DP, RB-like, E2F, and MuvB (DREAM) complex, which contains p130 and E2F4 or E2F5 [12,13,28,29,30,31,32,33]. pRB/E2F3b also contributes to suppression [7]. Upon growth stimulation, expression of cyclin D is induced, activating cyclin-dependent kinase (CDK) 4 and CDK6, which inactivate p130 by phosphorylation [13,34]. Repression of E2F target genes by the DREAM complex is relieved, and repressor E2Fs induce the expression of genes required for progression into the S phase, such as activator E2Fs, cyclin E and CDC6 [21,22,23,35,36]. Repressor E2Fs are exported into the cytoplasm by NES and replaced with induced activator E2Fs. In contrast to p130, which is inactivated by phosphorylation via CDK4/6, pRB is activated by mono-phosphorylation via CDK4/6 and regulates activity of activator E2Fs induced at the G1/S boundary [14,37]. Cyclin E activates CDK2, which inactivates pRB by hyper-phosphorylation [14], further activating E2F, along with expression of activator E2Fs. This positive feedback loop shifts RB inactivation and E2F activation from growth-stimulation dependent D-type CDK to E2F-dependent E-type CDK, thereby passing through the restriction point [38,39]. Indeed, analysis of the relationship between E2F activity and the restriction point revealed that once the E2F activity reached the requisite threshold, the cell cycle proceed autonomously, regardless of the presence of growth stimulation [40,41]. The *E2F7* and *E2F8* genes are targets of E2F and are induced at the G1/S boundary. E2F7 and E2F8 suppress E2F1 expression, thereby forming a negative feedback loop in the control of E2F1 activity [42]. At the end of S phase, activator E2Fs are degraded by the ubiquitin proteasome system [43,44,45]. In addition, activator E2Fs, with a cyclin A binding region, are phosphorylated by cyclin A/CDK2, which suppresses E2F activity by compromising the DNA binding activity of E2F [46,47].

In this manner, E2F plays central roles in G1 to S phase progression under the control of RB, and its activity is turned off to facilitate exit from the S phase.

## 4. Mechanism of Tumor Suppression Mediated by Deregulated E2F

E2F can also induce apoptosis and plays crucial roles in tumor suppression [2,4,5,48,49,50]. pRB function is disabled in almost all cancers by oncogenic changes, such as the deletion or mutation of pRB itself, the over-expression of cyclin/CDK, and/or the deletion of CDK inhibitors [51]. Consequently, enhanced E2F activity contributes to oncogenesis by activating growth-related genes to facilitate cell proliferation. However, in addition to growth-related genes, E2F freed from pRB control (deregulated E2F) also activates the Alternative reading frame (ARF), an upstream activator of the tumor suppressor p53, thereby linking the RB and p53 pathways, two major regulators of tumor suppression [49]. Among E2F family members, E2F1 is the most potent activator of pro-apoptotic tumor suppressor genes, and is a critical regulator of tumor suppression [48,52]. In normal cells, the expression of p53 is maintained at low levels by the mouse double minute 2 (MDM2 (HDM2 in human))-mediated ubiquitination and consequent proteasome-mediated degradation [8] (Figure 4A). ARF binds to and translocates MDM2 into the nucleolus, thereby suppressing p53 degradation by MDM2 in the nucleoplasm (Figure 4B) [8,53]. p53 plays central roles in tumor suppression by inducing cellular senescence or apoptosis [8,12,54]. p53 induces the expression of the CDK inhibitor p21^Cip1^ and inhibits CDK, thereby decreasing RB phosphorylation [8,12]. This leads to the activation of RB and inhibits E2F, suppressing cell cycle progression [12]. Therefore, the integrity of RB function is required for the induction of cellular senescence. However, in almost all cancers, RB function is disabled by oncogenic changes. In such a case, p53 induces apoptosis to protect cells from tumorigenesis. p53 induces apoptosis by activating a plethora of pro-apoptotic genes such as *Bax*, *Noxa* and *Apaf-1*.

Two major tumor suppressor pathways are the RB pathway and the p53 pathway. E2F plays crucial roles in tumor suppression by linking both pathways through activation of the *ARF* gene upon dysfunction of the RB pathway.

## 5. Roles of E2F in Tumorigenesis

Enhanced E2F activity observed in almost all cancers originates from alterations in the CDK-RB-E2F axis, and is a primary event in tumorigenesis, facilitating cell proliferation [7,51,55,56,57]. This is best exemplified by oncogenesis induced by oncogenes. Many oncogene products, such as Myc and those of oncogenic tumor viruses, including adenoviral E1a, E7 of the human papilloma virus (HPV), large T of SV40 and polyoma, and Tax of human T-cell leukemia virus type 1 (HTLV-1), converge on the CDK-RB-E2F axis to promote cell proliferation [58,59,60,61]. Deregulation of the CDK-RB-E2F pathway also causes genomic instability, aneuploidy, and centrosome amplification, thereby contributing to tumor promotion [57,62,63,64,65,66]. E2F also promotes the resistance of cancer cells to chemotherapy and radiation, cancer cell stemness, epithelial to mesenchymal transition (EMT), and metastasis [67]. Likely reflecting the roles or E2F in tumorigenesis and tumor progression, expression levels of E2Fs and E2F target genes are reported to correlate with the prognosis of cancer patients [68,69,70].

### 5.1. Promotion of Cell Proliferation

The promotion of cell proliferation is the primary event in tumorigenesis. E2F plays central roles in cell proliferation by activating a group of growth-related genes. Specifically, E2F is critical for passage through the restriction point, which controls cell cycle progression to later stages, by inducing *cyclin E* gene expression [35], which is necessary to proceed to the next cell cycle stage. Other target genes involved in cell cycle progression include *cyclin A*, *CDC2*, and *CDC25A* [71,72,73]. E2F target genes involved in S phase progression are those regulating the initiation of DNA replication, such as *Orc1*, *Cdc6*, *Mcms*, *Cdt1*, *Cdc45*, and *ASK* [36,38,74,75,76,77,78,79], and those involved in DNA synthesis, such as *DNA polymerase α*, *thymidylate synthase*, *proliferating cell nuclear antigen*, and *ribonucleotide reductase* [80]. E2F also regulates genes involved in checkpoints, such as *Chk1*, *Claspin*, *ATM*, and *BRCA1* [81,82,83,84], and those involved in DNA repair, such as *Mlh1* and *Rad51* [85]. Under certain cellular circumstance, repressor E2Fs also facilitate cell proliferation. For example, in hepatocellular carcinoma cells, E2F7 facilitates cell proliferation through the AKT/β-catenin/Cyclin D1 signaling pathway [86]. In mouse embryonic stem cells, E2F4 is free from RB family members and acts as a transcriptional activator for cell cycle genes, facilitating cell proliferation and survival [87].

Since E2F plays central roles in cell cycle progression and cell proliferation, suppressing enhanced E2F activity in cancer cells would be a fascinating approach to restrain the growth of cancer cells.

### 5.2. Maintenance of Cancer Cell Stemness

The initiation, metastasis, recurrence, and chemoresistance of many human cancers have been ascribed to cancer stem cells (CSCs) [88,89]. CSCs are cancer cells that possess characteristics associated with normal stem cells, especially the ability to give rise to all cell types found in a particular cancer sample. CSCs can maintain stem cell characteristics by undergoing asymmetric cell division, thereby generating one daughter cell with characteristics of CSCs and the other with those of a differentiated phenotype [90,91]. CSCs are therefore tumorigenic (tumor-initiating cells), and are the focus of interest regarding resistance to cancer therapy. Accumulating evidence indicates that E2F plays roles in the acquisition of the CSC phenotype [67,92]. E2F can directly activate the stem cell transcription factors. For example, NANOG is induced by toll-like receptor 4 (TLR4) signaling via the phosphorylation of E2F1, and the downregulation of NANOG slows down hepatocellular carcinoma (HCC) progression induced by alcohol, a Western diet, and hepatitis C virus protein in mice [93]. EIF5A2 positively regulates stemness in ovarian cancer cells by maintaining KLF4 expression through E2F1 [94]. Nicotine can induce the expression of Sox2, which is indispensable for self-renewal and maintenance of stem cell properties in non-small cell lung adenocarcinoma cells, through an AChR-Yap1-E2F1 signaling axis downstream of Src and Yes kinases [95]. In nicotine-induced proliferation of human non-small cell lung cancer cells, E2F1 can induce expression of stem cell factor (SCF), the ligand for c-Kit, facilitating the self-renewal of lung cancer side population cells [96]. MUC1-C (mucin 1 C terminal transmembrane subunit) could directly bind to E2F1, inducing the NOTCH1 signaling pathway and promoting NANOG expression, which in turn led to the promotion of CSCs stemness in prostate cancer stem cells [97]. ENPP1 (ectonucleotide pyrophosphatase/phosphodiesterase 1) is reported to induce characteristics of CSCs [98,99]. In addition, the knockdown of ENPP1 resulted in the decreased function of E2F1 and expression of target genes involved in G_1_/S transition, suggesting that ENPP1, by acting upstream of E2F1, is indispensable for the maintenance of CSCs [100]. The inhibition of E2F1 in chronic myeloid leukemia (CML) stem/progenitor cells (SPCs) reduced proliferation of CML SPCs, leading to p53-mediated apoptosis. In addition, E2F1 plays a pivotal role in regulating CML SPC proliferation status [101].

E2F also plays central roles in cell proliferation in CSCs. The proliferation of CSCs is well correlated with the ability of E2F to promote cell cycle progression, and E2F1-3a plays a central role in the G1/S phase transition by activating cell cycle-related target genes, as described above.

Since E2F regulates the development and maintenance of the CSC phenotype and its growth, suppressing E2F activity is expected to reduce stemness and the proliferation of CSCs. As CSCs play important roles in tumor initiation and metastatic colonization, suppressing E2F activity in CSCs would significantly contribute to cancer therapy.

### 5.3. Promotion of Epithelial-Mesenchymal Transition (EMT) and Metastasis

Epithelial-to-mesenchymal transition (EMT) is intimately related to cell motility, invasiveness, and metastasis, and also confers stemness, which facilitates metastatic colonization [102,103]. Accumulating evidence indicates involvement of E2F in the facilitation of EMT and metastasis [104]. For example, E2F1 promotes EMT by regulating *ZEB2* gene expression in small cell lung cancer [105], and E2F8 regulates proliferation and invasion through EMT in cervical cancer [106]. E2F7 facilitates EMT and metastasis in gallbladder carcinoma [107]. E2F6 supports the invasion and migration abilities of endometrial carcinoma cells [108].

CSCs have tumor initiating potential and exhibit higher invasive and metastatic ability compared to non-CSC populations, thereby playing major roles in metastasis [102,103]. E2F1 drives breast cancer metastasis by upregulating the *FGF13* gene and altering cell migration [109]. In the mouse model of metastatic breast cancer, interbreeding the mouse mammary tumor virus (MMTV)-polyomavirus middle T oncoprotein (PyMT) mice with *E2F1*, *E2F2*, or *E2F3* knockout mice revealed that E2F1 and E2F2 control the expression of genes critical to angiogenesis, remodeling of the extracellular matrix, tumor cell survival, and tumor cell interactions with vascular endothelial cells that facilitate metastasis to the lungs [110]. The analysis of long non-coding RNA *MYLK*-*AS1* revealed involvement of E2F7 in tumor progression and angiogenesis [111]. Similarly, the analysis of microRNA-302a/d revealed a role for E2F7 in self-renewal capability and cell cycle entry of liver cancer stem cells [86].

EMT plays important roles in metastasis, which is resistant to surgical treatment. Thus, suppressing E2F activity may reduce metastasis through suppression of EMT.

### 5.4. Chemoresistance

CSCs are the main culprit in the development of chemoresistance in cancer cells. Molecular mechanisms of chemoresistance include efficient drug efflux through ATP binding cassette (ABC) transporters, enhanced repair of DNA damage, and enhanced autophagy [112,113]. E2F1 promotes chemoresistance by inducing the expression of ABC transporter ABCG2 [114] and confers anticancer drug resistance by activating ABC transporter family members (*ABCA2* and *ABCA5*) and *Bcl-2* in malignant melanoma cells [115]. Similarly, E2F target genes modulating checkpoint and DNA repair are reported to be involved in chemoresistance. The E2F1-mediated upregulation of Rad51 contributes to the chemoresistance of prostate cancer cells under hypoxic conditions [116]. *BRCA1* expression is downregulated through E2F4 [117], and the downregulation of BRCA1 expression increases CSC-like populations in breast cancer cells [118]. Autophagy is a pro-survival process that allows cells to adapt to a variety of stressors. Blocking autophagy induced apoptosis and prevented colony formation in primary human acute myeloid leukemia cells [119]. A recently identified new member of the E2F3 family, E2F3d, lacks a DNA binding domain, due to alternative splicing, and localizes to the mitochondrial outer membrane, mediating hypoxia-induced autophagy of the mitochondria (mitophagy) [24].

The development of drug resistance is a major limitation to the efficacy of chemotherapy. Suppression of E2F activity may also contribute to cancer therapy by decreasing the occurrence of, or potentially reversing, chemoresistance.

## 6. Current Approaches Based on Enhanced E2F Activity in Cancer Cells

In almost all cancers, the RB pathway is compromised, and the E2F activity is enhanced, by oncogenic changes [8]. Enhanced E2F activity contributes to tumorigenesis by promoting cell proliferation, chemoresistance, stemness, metastasis, etc., as described above. Thus, enhanced E2F activity in cancer cells seems to be an attractive therapeutic tool to approach cancer treatment [120,121]. To specifically target cancer cells, several trials have been undertaken based on enhanced E2F activity in cancer cells.

### 6.1. Inhibition of Cancer Cell Proliferation by Suppressing Enhanced E2F Activity

CDK4/6 inhibitors are utilized, focusing on the CDK-RB-E2F axis to restrain the proliferation of cancer cells [122,123,124,125,126,127,128,129,130]. The inhibition of CDK activity reduces the phosphorylation of RB, thereby activating RB, suppressing E2F activity, and arresting cell cycle progression into the S phase. Among CDK4/6 inhibitors, palbociclib (PD0332991), ribociclib (LEE011), and abemaciclib (LY2835219) are approved by the Food and Drug Administration (FDA) for treatment of advanced breast cancers and are reported to suppress growth of tumors [131,132,133,134,135,136,137]. Fueled by the efficacy of these CDK4/6 antagonists, new CDK4/6 inhibitors are under development, such as PF-06873600, which inhibits CDK2 in addition to CDK4/6 [138]. However, the effects of CDK4/6 inhibitors are limited in tumors with deletion or mutation of pRB.

A small-molecule pan-E2F inhibitor HLM006474 and the nucleotide analogue ly101-4B are reported to suppress E2F activity. Treatment of multiple cancer cell lines with HLM006474 resulted in the downregulation of total E2F4 protein, as well as known E2F targets, along with a reduction in cell proliferation and an increase in apoptosis [139]. HLM006474 synergized with paclitaxel in a non-small cell lung carcinoma cell line [140]. Patient-derived xenografts of pancreatic ductal adenocarcinoma were grouped into E2F-highly dependent and E2F-lowly dependent cells by bioinformatics-based analysis and were subsequently tested for sensitivity to ly101-4B. E2F-higly dependent cells were more sensitive to the treatment [69].

Several peptides designed to block E2F binding to target DNA showed the ability to suppress tumor cell growth. The introduction of peptides that functionally antagonize E2F DNA binding activity into mammalian tumor cells caused the rapid onset of apoptosis, an outcome that correlates with the inactivation of physiological E2F [141]. A 7-mer peptide, which binds to the E2F-1 consensus sequence, was coupled with penetratin to enhance cellular uptake (penetratin-peptide (PEP)). PEP demonstrated potent in vitro cytotoxic effects against a range of cancer cell lines. PEP inhibited the transcription of E2F-1 and also several important E2F-regulated enzymes involved in DNA synthesis. The treatment of mice bearing the human small-cell lung carcinoma H-69 with the PEP encapsulated in PEGylated liposomes (PL-PEP) caused tumor regression, without significant toxicity [142,143]. D-Arg penetratin peptide (D-Arg PEP) was both more stable and more cytotoxic compared to the L-Arg PEP [144]. Branched tetravalent molecules of two peptides that specifically recognize E2F and inhibit the DNA binding of E2F/DP heterodimers were fused to a cell-penetrating peptide derived from the HTV-Tat protein. The incubation of human tumor cells with these branched Tat-containing peptides led to an inhibition of cell proliferation and an induction of apoptosis [145].

These results support that notion that the inhibition of enhanced E2F activity in tumor cells is a promising strategy to restrain the proliferation of cancer cells. However, there is a possibility that this approach also inhibits the growth of normal cells, since E2F activity is required for normal cell proliferation.

### 6.2. Utilizing Enhanced E2F Activity to Specifically Target Cancer Cells

Trials have been undertaken to enable the replication of recombinant viruses, depending on enhanced E2F activity in cancer cells. For example, driving E1a and E1b required for the replication of adenovirus by the E2F-responsive promoter enabled the replication of the virus in cancer cells with high E2F activity and effectively lysed the cells [146,147,148,149]. This approach is currently the subject of ongoing clinical trials [150] and has been improved by its combination with other mechanistic, therapeutic targets, such as hyaluronidase and immune activation [151,152,153,154,155,156]. As an E2F-responsive promoter, the E2F1 promoter has been employed in such studies. However, since the *E2F1* gene is a growth-related E2F target, and the promoter is also activated by growth stimulation in normal cells [21], this approach may also adversely affect normal growing cells.

## 7. Targeting Deregulated E2F1 Activity Unique to Cancer Cells

The approach using the growth-related E2F target promoter may negatively impact normal growing cells. This is because growth-related E2F targets are activated by physiological E2F activity, induced by growth stimulation, in normal cells. To overcome this problem, the promoter of the tumor suppressor *ARF* gene shows great potential, as this promoter is specifically activated by deregulated E2F activity that is only present in cancer cells [9].

In cancer cells, the impairment of the RB pathway leads to dysfunction of pRB, generating deregulated E2F activity. In addition, the ARF-p53 pathway is also impaired, and the induction of apoptosis is compromised. Therefore, cancer cells harbor deregulated E2F1 activity, which activates the *ARF* gene. Importantly, E2F activity induced by growth stimulation does not activate the *ARF* gene [9]. Therefore, normal growing cells do not harbor deregulated E2F activity, which activates the *ARF* gene. Together, deregulated E2F activity, and the consequent activation of *ARF* gene expression, are unique, targetable characteristics of cancer cells [9].

To explain the E2F regulation of pro-apoptotic genes, such as *ARF*, the threshold model has been proposed [1]. According to this model, when the total amount of E2F, free from RB, reaches an initial threshold, the growth-related genes are activated. When total free E2F reaches a second higher threshold, pro-apoptotic genes, in addition to growth-related genes, are activated. However, conflicting results with respect to this model have been reported. E2F requires heterodimeric partner DP to bind to in order to activate growth-related genes. In response to the knockdown of DP expression in normal human fibroblasts, the activation of growth-related genes by overexpressed E2F1 was compromised. However, the activation of the *ARF* genes was unaffected by the loss of DP. This argues against the threshold model, and indicates that E2F activity, which activates the *ARF* gene, is not dependent on DP, instead constituting a distinct activity from that which activates growth-related genes [157]. Therefore, deregulated E2F activity, activating the *ARF* gene, is distinct from simply enhanced E2F activity in cancer cells. It is expected that targeting cancer-cell specific deregulated E2F and the resulting enhanced ARF promoter activity will increase the cancer cell specificity of such treatments.

### 7.1. To Specifically Target Cancer Cells Utilizing Deregulated E2F Activity

It is reported that the ARF promoter exhibits more cancer-cell specific activity than the previously utilized E2F1 promoter [158]. Activities of E2F1 and ARF promoters were compared between normal human fibroblasts and several cancer cell lines. In all of the cancer cell lines tested, ARF promoter activity showed higher cancer-cell specificity than the E2F1 promoter. Recombinant adenoviruses, in which the herpes simplex virus *thymidine kinase* (*HSV-TK*) gene is driven by ARF or the E2F1 promoter (ARF-TK or E2F1-TK), were generated. Normal human fibroblasts and several cancer cell lines were infected with the viruses, and in the presence of ganciclovir, cytotoxicity was compared between ARF and the E2F1 promoter. Ganciclovir is the inactive precursor, which when metabolized in cells expressing HSV-TK, is converted into an active anti-cancer agent to kill cells. Using this system, ARF-TK was a more specific inducer of cell death in cancer cells compared to E2F1-KT [158]: ARF-TK showed lower cytotoxicity than E2F1-TK in normal human fibroblasts, but comparable cytotoxicity to E2F1-TK in cancer cells. These results suggest that utilizing deregulated E2F activity, unique to cancer cells, to activate an ARF promoter-driven TK construct, will enable cancer-cell specific ganciclovir-mediated cell death, while preserving normal growing cells. These results also support the notion that deregulated E2F activity would be a better therapeutic tool than simply enhanced activity to specifically target cancer cells.

### 7.2. To Enhance Deregulated E2F Activity Specifically Restricted to Cancer Cells

Deregulated E2F activates a group of tumor suppressor genes in addition to the *ARF* gene [10,11,159]. TAp73, a p53 family member, can activate p53 target genes independent of p53. The *TAp73* gene is an E2F target that is specifically activated by deregulated E2F [10,160,161]. Deregulated E2F also activates the *p27^Kip1^* gene, which is not activated by E2F induced in response to growth stimulation [159]. p27^Kip1^ inhibits CDK, thereby activating RB and restraining cell cycle progression. Screening for genes specifically activated by deregulated E2F activity induced by the overexpression of E2F1 or the forced inactivation of RB by adenovirus E1a in normal human fibroblasts identified 9 novel genes (*Bim*, *Aspp1*, *RASSF1*, *JMY*, *MOAP1*, *RBM38*, *ABTB1*, *RBBP4* and *RBBP7*) [11]. Thus, it is expected that deregulated E2F can induce cellular senescence or apoptosis, independent of p53, via alternate pathways mediated by TAp73, p27^Kip1^, Bim, and other such gene products. In fact, the overexpression of E2F1 in U-2 OS cells, in which the *ARF* gene is silenced by DNA methylation, induces apoptosis (our unpublished observation). In addition, *ARF* knockout did not alleviate apoptosis induced by *RB1* knockout, indicating that an alternate pathway(s), independent of ARF, is important for the induction of apoptosis upon the loss of pRB function in vivo [162]. These results indicate that there are other pathways by which deregulated E2F can induce apoptosis (Figure 5). In spite of this, cancer cells survive with deregulated E2F activity, implying that deregulated E2F1 activity may be suppressed by yet unknown factors. If we could increase deregulated E2F1 activity in cancer cells, we might be able to induce apoptosis through p53-independent pathways, in which the ARF-p53 pathway is disabled. Thus, increasing deregulated E2F1 activity in cancer cells represents a novel, cancer-specific therapeutic approach [163].

CDK inhibitors are generally used for cancer therapy [131,132,133,134]. It is reported that in tissue culture cells, the over-expression of CDK inhibitor p21^Cip1^ enhanced deregulated E2F activity and increased cytotoxic gene expression driven by the ARF promoter in cancer cells [164]. This suggests that the effects of CDK inhibitors are mediated, at least in part, by increasing deregulated E2F1 activity, supporting the proposed treatment strategy of enhancing deregulated E2F activity in cancer cells, specifically inducing apoptosis.

## 8. Conclusions

Since promotion of cell proliferation is a primary event in tumorigenesis, a defective RB pathway and the consequent enhancement of E2F activity are observed in almost all cancers. Thus, enhanced E2F activity is a fascinating tool to specifically target cancer cells. However, E2F activity is also increased in normal growing cells, since growth stimulation inactivates RB and activates E2F. Hence, targeting enhanced E2F activity that activates growth-related genes has the potential to negatively impact normal growing cells. In contrast, deregulated E2F activity, which activates tumor suppressor genes, is unique to cancer cells, since enhanced E2F activity, in response to physiologic growth stimulation, does not activate these genes in normal growing cells. Therefore, deregulated E2F activity that specifically activates tumor suppressor genes has inherent advantages over enhanced E2F activity as a cancer-specific therapeutic tool.

## 9. Future Perspectives

Accumulating evidence points to the roles of enhanced E2F activity in tumorigenesis by facilitating cell proliferation, chemoresistance, stemness, metastasis, and other oncogenic processes. Thus, E2F is an attractive therapeutic tool with which to target cancer cells. However, suppressing E2F activity may potentially adversely affect normal growing cells that are dependent on E2F activity for cell proliferation.

Based on the mechanism of tumorigenesis, deregulated E2F activity is a specific characteristic of cancer cells, and as such, it is a fascinating potential therapeutic target. An important consideration is how to efficiently utilize deregulated E2F activity to damage cancer cells by expressing a cytotoxic gene. Among genes specifically activated by deregulated E2F, several tumor suppressor genes have been identified, such as *ARF*, *TAp73*, and 9 others [9,10,11], in some of which, E2F responsive elements have been characterized [9,10,11]. It is known that the multimerization of responsive elements increases the responsiveness to the transcription factors. Thus, it may be possible to increase the cancer-cell specificity of cytotoxic genes by combining highly active, concatenated E2F-responsive elements upstream of a core promoter that exhibits low basal activity in normal cells.

CSCs are the main culprit, as they confer chemoresistance to cancer cells, leading to recurrence after treatment. Thus, it is ideal to target CSCs for the radical treatment of cancer. We now know that deregulated E2F activity is a cancer-cell specific characteristic, but we are not certain whether it also exists in CSCs. If it exists in CSCs, it would be a valuable tool to target. Whether deregulated E2F activity also exists in CSCs remains to be confirmed. 

Deregulated E2F can induce apoptosis through pathways other than ARF-p53. However, cancer cells survive with deregulated E2F activity. This suggests that, in cancer cells, some factors may suppress deregulated E2F activity. Conversely, there may be factors which enhance deregulated E2F activity in normal cells, but inactivate this activity in cancer cells. Thus, it is important to elucidate factors that modulate deregulated E2F1 activity. This would enable us to enhance deregulated E2F1 activity by eliminating or attenuating the actions of suppressors and/or enhancing those of activating factors. For example, the development of small molecules which disrupt the interaction of repressive factors with deregulated E2F1 in cancer cells would be an effective cancer-specific treatment. In this context, it is crucial to elucidate the biochemical nature of, and regulatory mechanisms underlying, the actions of deregulated E2F1 as a potential highly specific therapeutic agent in cancer treatment.

## Figures and Tables

**Figure 1 genes-14-00393-f001:**
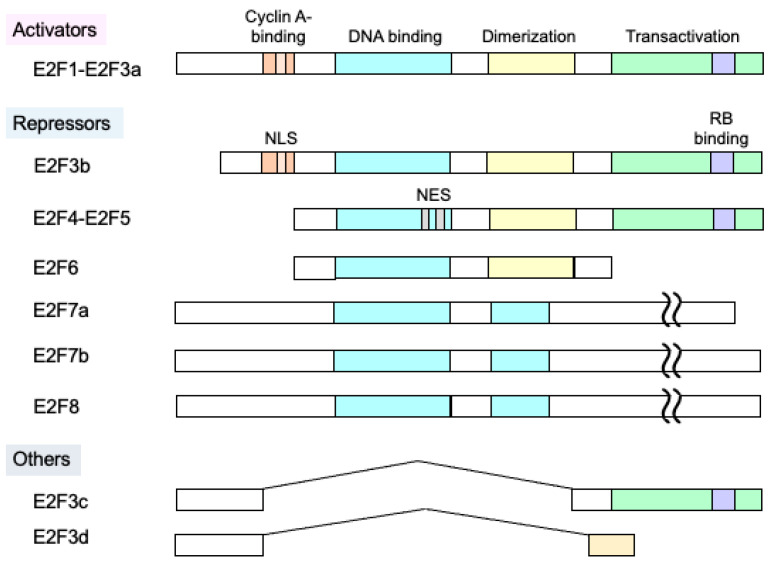
Structure of E2F family members. There are eight E2F family members (E2F1-E2F8), which are divided into activator E2Fs (E2F1~E2F3a) and repressor E2Fs (E2F3b~E2F8). Activator E2Fs are mainly involved in the induction of target gene expression at the G1/S boundary, free from RB. Repressor E2Fs are mainly involved in the repression of target gene expression, together with RB in quiescent cells. E2F6-E2F8 lack an RB binding domain and repress target gene expression independent of RB.

**Figure 2 genes-14-00393-f002:**
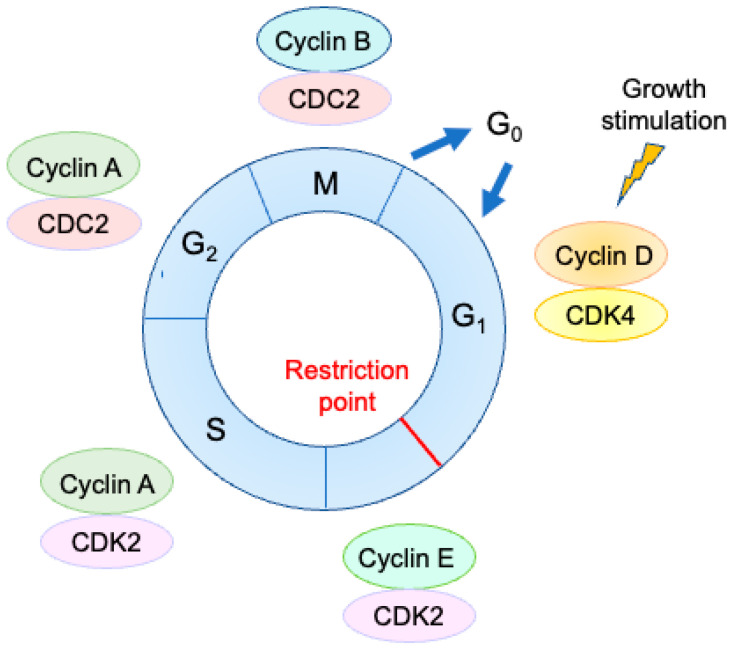
Cell cycle regulation by cyclins and CDKs. Each stage of the cell cycle is governed by specific cyclin/CDK complexes. Progression of the cell cycle to the restriction point, located in late G1, is dependent on growth stimulation. After passing through the restriction point, cell cycle progression proceeds autonomously through the end of mitosis.

**Figure 3 genes-14-00393-f003:**
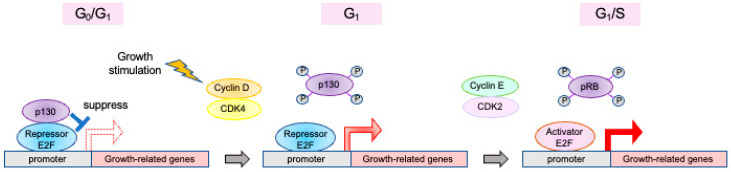
Mechanism of cell cycle progression mediated by E2F. In the resting state, the DREAM complex, including repressor E2Fs and p130, actively suppresses E2F target gene expression. Upon growth stimulation, the expression of cyclin D is induced, activating CDK4/6, which phosphorylates p130 and relieves repression by the DREAM complex. Repressor E2Fs, freed from repression by the DREAM complex, now induces the expression of growth-related genes such as cyclin E and activator E2Fs, further inactivating pRB and activating E2F. This feedback loop shifts RB inactivation and E2F activation from the growth-dependent D-type CDK to the E2F-dependent E-type CDK, thereby facilitating transit through the restriction point. After the restriction point, progression of the cell cycle proceeds autonomously to the end of mitosis.

**Figure 4 genes-14-00393-f004:**
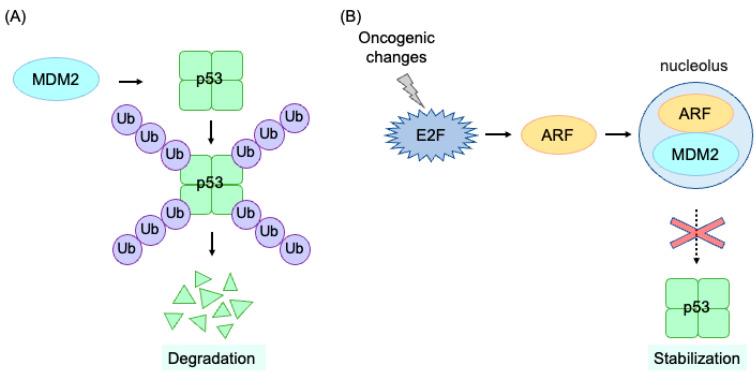
Regulatory mechanism of p53 activity by protein degradation and stabilization. (**A**) In normal cells, the level of p53 protein is kept low by MDM2-mediated ubiquitination and proteasome-mediated degradation. (**B**) Upon oncogenic changes, E2F is activated independently of pRB control (deregulated E2F) and induces expression of the tumor suppressor gene *ARF*. ARF binds to and translocates MDM2 into the nucleolus, thereby preventing MDM2-mediated ubiquitination and degradation and stabilizing p53 protein.

**Figure 5 genes-14-00393-f005:**
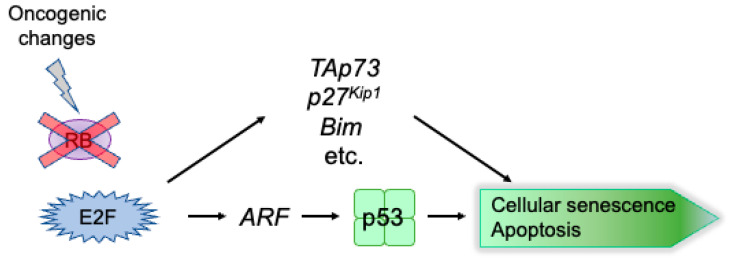
Multiple pathways for E2F-mediated tumor suppression. Deregulated E2F induces cellular senescence or apoptosis through multiple pathways. One is mediated through p53 by activating the tumor suppressor gene *ARF*. Others are mediated through *TAp73*, *Bim*, and other tumor suppressor genes. Many cancer cells survive by dysfunction of the ARF-p53 pathway, despite the presence of other tumor suppressive pathways.

## Data Availability

Not applicable.

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
