# Peer review of "Deregulated E2F Activity as a Cancer-Cell Specific Therapeutic Tool"

_genes, 2023, doi:10.3390/genes14020393_

Round 1

Reviewer 1 Report

This manuscript is from a well-respected expert in the E2F field and focuses on the role of deregulated E2F activity in driving cancer phenotypes and in providing various therapeutic opportunities. It begins with a comprehensive review of the E2F family and its role in regulating cell proliferation during the cell cycle. This includes recent findings on newly recognized E2F3 isoforms. Although much of the information in these sections is well-established, it also includes a few dogmatic concepts that are outdated. For instance, it is now known that cyclin D-CDK4/6 complexes only monophosphorylate RB and that these monophosphorylated forms of RB remain active (i.e., still bind E2F), as opposed to what is indicated in the text and Figure 3. The sections on the role of E2F in cancer stem cell maintenance and promotion of EMT are nice syntheses of older and more recent findings. A unique aspect of this review is the thoughtful discussion of how deregulated E2F activity can be used to specifically target cancer cells. Below are a few suggestions that may further improve the manuscript.

1.      The title of the manuscript specifies deregulated E2F1 activity but most of the review refers to the general deregulation of multiple E2F family members due to inactivation or loss of RB.

2.      Although the ability of E2F1 to suppress tumor development is well established from mouse knockout studies, the role of apoptosis as the mechanism of tumor suppression is controversial. In at least some animal model studies, tumor suppression by E2F1 was more closely associated with senescence or maintaining genome integrity. Can the authors provide references from the primary literature to support their model (Figure 5) that E2F suppresses tumor development by inducing ARF-mediated apoptosis in response to RB inactivation?

3.      The finding that E2F physically associates with the ARF promoter independent of DP is quite interesting. Can the authors add further details on the mechanism, such as whether monomeric E2F directly binds DNA or whether there is another binding partner that assists E2F in binding?

Author Response

We appreciate valuable comments. According to the comments, we extensively revised our manuscript. Our point by point answers to the comments are listed below.  

For instance, it is now known that cyclin D-CDK4/6 complexes only monophosphorylate RB and that these monophosphorylated forms of RB remain active (i.e., still bind E2F), as opposed to what is indicated in the text and Figure 3.

A: We are sorry for the confusion, which was caused by our way of writing to avoid complexity. In this manuscript, RB refers to RB family members including pRB, p130 and p107 as explained in page 2, line 94-95 “the tumor suppressor pRB and its family members p107 and p130 (collectively referred to RB)”. The indicated point was specifically regarding pRB, whereas we have been implicating p130 at this situation. To avoid confusion, we discriminated between p130 and pRB and explained in detail in Page 4 lines 160 to 171 and in Figure 3.

  1. The title of the manuscript specifies deregulated E2F1 activity but most of the review refers to the general deregulation of multiple E2F family members due to inactivation or loss of RB.

A: We appreciate the comment. We changed E2F1 to E2F in the title.

  1. Although the ability of E2F1 to suppress tumor development is well established from mouse knockout studies, the role of apoptosis as the mechanism of tumor suppression is controversial. In at least some animal model studies, tumor suppression by E2F1 was more closely associated with senescence or maintaining genome integrity. Can the authors provide references from the primary literature to support their model (Figure 5) that E2F suppresses tumor development by inducing ARF-mediated apoptosis in response to RB inactivation?

A: We appreciate the comment for in vivo situation. Figure 5 was just to show that there are multiple pathways for induction of apoptosis by deregulated E2F1, which is suggested mainly by in vitro experiments. As not only apoptosis but also cellular senescence are observed in pre-malignant tissues in vivo, we think that induction of cellular senescence is also important for E2F1-mediated tumor suppression in vivo. So we refrained from focusing on apoptosis and added cellular senescence. Accordingly, we modified the manuscript where applicable.

  1. The finding that E2F physically associates with the ARF promoter independent of DP is quite interesting. Can the authors add further details on the mechanism, such as whether monomeric E2F directly binds DNA or whether there is another binding partner that assists E2F in binding?

A: We are currently working on the points. We are speculating that E2F1 forms homodimer to bind to tumor suppressor gene promoters. This is based on our observation that, using bimolecular fluorescence complementation (BiFC) method, N-terminal Venus-fused E2F1 and C-terminal Venus-fused E2F1 generated Venus fluorescence in the nucleus, suggesting that deregulated E2F1 can form homodimer to bind to target promoters. But this is unpublished result and we think it would be better not to include in this review manuscript. We are also speculating that there are other factors, which assist homodimeric E2F1 to bind to target promoters. This is based on our observation that, among novel E2F1 interacting factors that we identified, DDX5 and WDR1 enhance E2F1 activation of the tumor suppressor genes. This suggests that there are factors, which support deregulated E2F1 to bind to target promoters, but not necessarily as a heterodimeric partner of E2F1. But this is also unpublished result and we think it would be better not to include in review manuscript.

Reviewer 2 Report

Deregulated E2F1 activity as a cancer cell specific therapeutic tool

By: Rinka Nakajima et. al.

The manuscript does not comply with the journal's submission format, as indicated in item 2 of the Genes MDPI checklist.

  1. (Use the Microsoft Word template or LaTeX template to prepare your manuscript)
  2.  

Its important to classify the article  (revision, original, etc) to avoid confusion

Suggested to update the bibliography; only 20% (26 of 129) are within the 5-year period, taking into account 2022 as a reference.

Suggested to highlight the conclusion

Similar articles in the field have to be cited and describe how different this article is.

In each of the subtopics, add discussion, compare with other authors, as it seems a description only a collection of information.

Author Response

We appreciate valuable comments. According to the comments, we extensively revised our manuscript. Our point by point answers to the comments are listed below.

The manuscript does not comply with the journal's submission format, as indicated in item 2 of the Genes MDPI checklist.

  1. Use the Microsoft Word template or LaTeX template to prepare your manuscript.

A: We are sorry to have overlooked the templates as we saw the description “Genes now accepts free format submission” under “Free Format Submission” in “Instructions for Authors”. The publisher kindly put our manuscript into the template and our manuscript is now in the Microsoft Word template.

  1. It’s important to classify the article  (revision, original, etc) to avoid confusion.

A: This is now a revised version of a review article.

Suggested to update the bibliography; only 20% (26 of 129) are within the 5-year period, taking into account 2022 as a reference.

A: We tried to update the bibliography as much as possible.

Suggested to highlight the conclusion.

A: We added “Conclusion” and strengthened that deregulated E2F activity is a useful tool for cancer cell specific therapeutic approach. Page 11 line 988 to page 12 line 1117.

Similar articles in the field have to be cited and describe how different this article is.

A: We cited general reviews of E2F and described how different from the general reviews. Page 2 lines 82 to 86.

In each of the subtopics, add discussion, compare with other authors, as it seems a description only a collection of information.

A: We added discussion at each of the subtopics where not yet added. Page 4 lines 183-184, page 3 lines 237-239, page 7 lines 425-427, lines 462-465, page 8 lines 623-624, lines 642-644, page 10 lines 961-963.

Round 2

Reviewer 2 Report

Accept in present form